# SCALABLE MODEL COMPRESSION BY ENTROPY PENALIZED REPARAMETERIZATION

**Deniz Oktay**[*]
Princeton University
Princeton, NJ, USA
`doktay@cs.princeton.edu`

**Johannes Ballé**
Google Research
Mountain View, CA, USA
`jballe@google.com`

**Saurabh Singh**
Google Research
Mountain View, CA, USA
`saurabhsingh@google.com`

**Abhinav Shrivastava**
University of Maryland, College Park
College Park, MD, USA
`abhinav@cs.umd.edu`

## ABSTRACT

We describe a simple and general neural network weight compression approach, in which the network parameters (weights and biases) are represented in a "latent" space, amounting to a reparameterization. This space is equipped with a learned probability model, which is used to impose an entropy penalty on the parameter representation during training, and to compress the representation using a simple arithmetic coder after training. Classification accuracy and model compressibility is maximized jointly, with the bitrate–accuracy trade-off specified by a hyperparameter. We evaluate the method on the MNIST, CIFAR-10 and ImageNet classification benchmarks using six distinct model architectures. Our results show that state-of-the-art model compression can be achieved in a scalable and general way without requiring complex procedures such as multi-stage training.

## 1 INTRODUCTION

Artificial neural networks (ANNs) have proven to be highly successful on a variety of tasks, and as a result, there is an increasing interest in their practical deployment. However, ANN parameters tend to require a large amount of space compared to manually designed algorithms. This can be problematic, for instance, when deploying models onto devices over the air, where the bottleneck is often network speed, or onto devices holding many stored models, with only few used at a time. To make these models more practical, several authors have proposed to compress model parameters (Han et al., 2016; Louizos, Ullrich, et al., 2017; Molchanov et al., 2017; Havasi et al., 2019). While other desiderata often exist, such as minimizing the number of layers or filters of the network, we focus here simply on model compression algorithms that 1. minimize compressed size while maintaining an acceptable classification accuracy, 2. are conceptually simple and easy to implement, and 3. can be scaled easily to large models.

Classical data compression in a Shannon sense (Shannon, 1948) requires discrete-valued data (i.e., the data can only take on a countable number of states) and a probability model on that data known to both sender and receiver. Practical compression algorithms are often *lossy*, and consist of two steps. First, the data is subjected to (re-)quantization. Then, a Shannon-style *entropy coding* method such as arithmetic coding (Rissanen and Langdon, 1981) is applied to the discrete values, bringing them into a binary representation which can be easily stored or transmitted. Shannon's source coding theorem establishes the entropy of the discrete representation as a lower bound on the average length of this binary sequence (the *bit rate*), and arithmetic coding achieves this bound asymptotically. Thus, entropy is an excellent proxy for the expected model size.

The type of quantization scheme affects both the fidelity of the representation (in this case, the precision of the model parameters, which in turn affects the prediction accuracy) as well as the bit

---

[*]Work performed during the Google AI Residency Program. (http://g.co/airesidency)

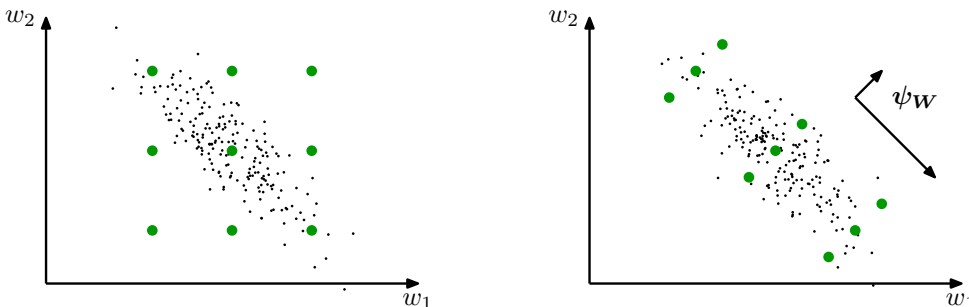

Figure 1: Visualization of representers in scalar quantization vs. reparameterized quantization. The axes represent two different model parameters (e.g., linear filter coefficients). Dots are samples of the model parameters, discs are the representers. Left: in scalar quantization, the representers must be given by a Kronecker product of scalar representers along the cardinal axes, even though the distribution of samples may be skewed. Right: in reparameterized scalar quantization, the representers are still given by a Kronecker product, but in a transformed (here, rotated) space. This allows a better adaptation of the representers to the parameter distribution.

rate, since a reduced number of states coincides with reduced entropy. ANN parameters are typically represented as floating point numbers. While these technically have a finite (but large) number of states, the best results in terms of both accuracy and bit rate are typically achieved for a significantly reduced number of states. Existing approaches to model compression often acknowledge this by quantizing each individual linear filter coefficient in an ANN to a small number of pre-determined values (Louizos, Reisser, et al., 2019; Baskin et al., 2018; F. Li et al., 2016). This is known as scalar quantization (SQ). Other methods explore vector quantization (VQ), closely related to $k$-means clustering, in which each vector of filter coefficients is quantized jointly (Chen, J. Wilson, et al., 2015; Ullrich et al., 2017). This is equivalent to enumerating a finite set of representers (representable vectors), while in SQ the set of representers is given by the Kronecker product of representable scalar elements. VQ is much more general than SQ, in the sense that representers can be placed arbitrarily: if the set of useful filter vectors all live in a subregion of the entire space, there is no benefit in having representers outside of that region, which may be unavoidable with SQ (fig. 1, left). Thus, VQ has the potential to yield better results, but it also suffers from the "curse of dimensionality": the number of necessary states grows exponentially with the number of dimensions, making it computationally infeasible to enumerate them explicitly, hence limiting VQ to only a handful of dimensions in practice. One of the key insights leading to this paper is that the strengths of SQ and VQ can be combined by representing the data in a "latent" space. This space can be an arbitrary rescaling, rotation, or otherwise warping of the original data space. SQ in this space, while making quantization computationally feasible, can provide substantially more flexibility in the choice of representers compared to the SQ in the data space (fig. 1, right). This is in analogy to recent image compression methods based on autoencoders (Ballé, Laparra, et al., 2017; Theis et al., 2017).

The contribution of this paper is two-fold. First, we propose a novel end-to-end trainable model compression method that uses scalar quantization and entropy penalization in a *reparameterized* space of model parameters. The reparameterization allows us to use efficient SQ, while achieving flexibility in representing the model parameters. Second, we provide state-of-the-art results on a variety of network architectures on several datasets. This demonstrates that more complicated strategies involving pretraining, multi-stage training, sparsification, adaptive coding, etc., as employed by many previous methods, are not necessary to achieve good performance. Our method scales to modern large image datasets and neural network architectures such as ResNet-50 on ImageNet.

## 2 ENTROPY PENALIZED REPARAMETERIZATION

We consider the classification setup, where we are given a dataset $D = \{(\boldsymbol{x}_1, y_1), ...(\boldsymbol{x}_N, y_N)\}$ consisting of pairs of examples $\boldsymbol{x}_i$ and corresponding labels $y_i$. We wish to minimize the expected negative log-likelihood on $D$, or cross-entropy classification loss, over $\Theta$, the set of model parame-

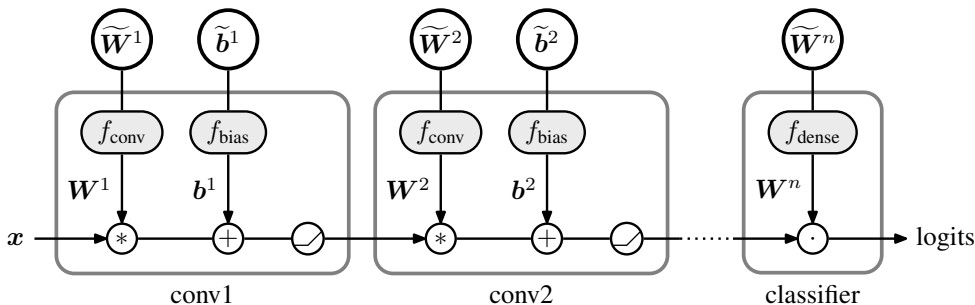

Figure 2: Classifier architecture. The $\Phi$ tensors (annotated with a tilde) are stored in their compressed form. During inference, they are read from storage, uncompressed, and transformed via $f$ into $\Theta$, the usual parameters of a convolutional or dense layer (denoted without a tilde).

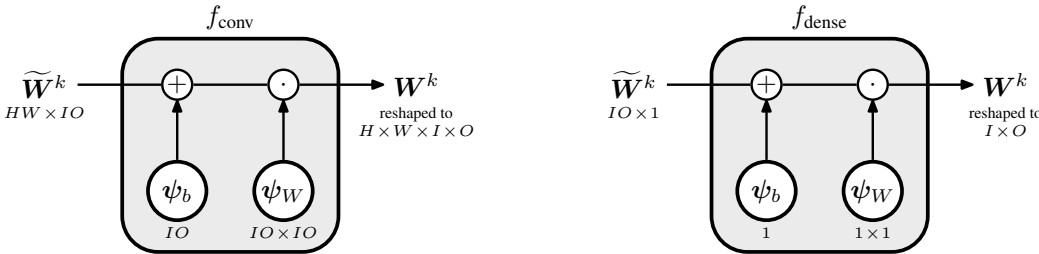

Figure 3: The internals of $f_{\text{conv}}$ and $f_{\text{dense}}$ in our experiments for layer $k$, annotated with the dimensionalities. In $f_{\text{conv}}$, $H$, $W$, $I$, $O$ refer to the convolutional height, width, input channel, output channel, respectively. For $f_{\text{dense}}$, $I$ and $O$ refer to the number of input and output activations. For $f_{\text{conv}}$, we use an affine transform, while for $f_{\text{dense}}$ we use a scalar shift and scale, whose parameters are captured in $\Psi$. Note that in both cases, the number of parameters of $f$ itself (labeled as $\psi$) is significantly smaller than the size of the model parameters it decodes.

ters:

$$\Theta^* = \arg\min_{\Theta} \sum_{(\boldsymbol{x}, y) \sim D} -\log p(y \mid \boldsymbol{x}; \Theta), \tag{1}$$

where $p(y \mid \boldsymbol{x}; \Theta)$ is the likelihood our model assigns to a dataset sample $(\boldsymbol{x}, y)$. The likelihood function is implemented using an ANN with parameters $\Theta = \{\boldsymbol{W}^1, \boldsymbol{b}^1, \boldsymbol{W}^2, \boldsymbol{b}^2, \ldots, \boldsymbol{W}^N\}$, where $\boldsymbol{W}^k$ and $\boldsymbol{b}^k$ denote the weight (including convolutional) and bias terms at layer $k$, respectively.

Compressing the model amounts to compressing each parameter in the set $\Theta$. Instead of compressing each parameter directly, we compress reparameterized forms of them. To be precise, we introduce the reparameterizations $\Phi = \{\widetilde{\boldsymbol{W}}^1, \widetilde{\boldsymbol{b}}^1, \widetilde{\boldsymbol{W}}^2, \widetilde{\boldsymbol{b}}^2, \ldots, \widetilde{\boldsymbol{W}}^N\}$ and parameter decoders $f_{\text{conv}}$, $f_{\text{dense}}$, $f_{\text{bias}}$ such that

$$\boldsymbol{W}^k = f_{\text{conv}}\big(\widetilde{\boldsymbol{W}}^k\big) \qquad \text{if layer } k \text{ is convolutional}, \tag{2}$$

$$\boldsymbol{W}^k = f_{\text{dense}}\big(\widetilde{\boldsymbol{W}}^k\big) \qquad \text{if layer } k \text{ is fully connected}, \tag{3}$$

$$\boldsymbol{b}^k = f_{\text{bias}}\big(\widetilde{\boldsymbol{b}}^k\big) \qquad \text{if layer } k \text{ has a bias}. \tag{4}$$

We can think of each parameter decoder $f$ as a mapping from reparameterization space to parameter space. For ease of notation, we write $\mathcal{F} = \{f_{\text{conv}}, f_{\text{dense}}, f_{\text{bias}}\}$ and $\Theta = \mathcal{F}(\Phi)$. The parameter decoders themselves may have learnable parameters, which we denote $\Psi$. Our method is visually summarized in figs. 2 and 3.

## 2.1 COMPRESSING $\Phi$ WITH ENTROPY CODING

In order to apply a Shannon-style entropy coder efficiently to the reparameterizations $\Phi$, we need a discrete alphabet of representers and associated probabilities for each representer. Rather than handling an expressive set of representers, as in VQ, we choose to fix them to the integers, and

achieve expressivity via the parameter decoders $\mathcal{F}$ instead.

Each reparameterization $\phi \in \Phi$ (i.e. a $\widetilde{\boldsymbol{W}}$ or $\widetilde{\boldsymbol{b}}$ representing a weight or bias, respectively) is a matrix in $\mathbb{Z}^{d \times \ell}$ interpreted as consisting of $d$ samples from a discrete probability distribution producing vectors of dimension $\ell$. We fit a factorized probability model

$$q(\boldsymbol{\phi}) = \prod_{j=1}^{d} \prod_{i=1}^{\ell} q_i(\phi_{j,i}) \tag{5}$$

to each column $i$ of $\boldsymbol{\phi}$, using $\ell$ different probability models $q_i$ for each corresponding parameter decoder (the form of $q_i$ is described in the next section). Fitting of probability models is often done by minimizing the negative log-likelihood. Assuming $\phi$ follows the distribution $q$, Shannon's source coding theorem states that the minimal length of a bit sequence encoding $\phi$ is the self-information of $\phi$ under $q$:

$$I(\boldsymbol{\phi}) = -\log_2 q(\boldsymbol{\phi}), \tag{6}$$

which is identical to Shannon cross entropy up to an expectation operator, and identical to the negative log likelihood up to a constant factor. By minimizing $I$ over $q$ and $\phi$ during training, we thus achieve two goals: 1) we fit $q$ to the model parameters in a maximum likelihood sense, and 2) we directly optimize the parameters for compressibility.

After training, we design an arithmetic code for $q$, and use it to compress the model parameters. This method incurs only a small overhead over the theoretical bound due to the finite length of the bit sequence (arithmetic coding is asymptotically optimal). Practically, the overhead amounts to less than 1% of the size of the bit sequence; thus, self-information is an excellent proxy for model size. Further overhead results from including a description of $\Psi$, the parameters of the parameter decoders, as well as of $q$ itself (in the form of a table) in the model size. However, these can be considered constant and small compared to the total model size, and thus do not need to be explicitly optimized for.

The overall loss function is simply the additive combination of the original cross-entropy classification loss under reparameterization with the self-information of all reparameterizations:

$$L(\Phi, \Psi) = \sum_{(\boldsymbol{x},y) \sim D} -\log p\big(y \mid \boldsymbol{x}; \mathcal{F}(\Phi)\big) + \lambda \sum_{\boldsymbol{\phi} \in \Phi} I(\boldsymbol{\phi}). \tag{7}$$

We refer to the second term (excluding the constant $\lambda$) as the *rate loss*. By varying $\lambda$ across different experiments, we can explore the Pareto frontier of compressed model size vs. model accuracy. To compare our method to other work, we varied $\lambda$ such that our method produced similar accuracy, and then compared the resulting model size.

## 2.2 DISCRETE OPTIMIZATION

Since $\Phi$ is discrete-valued, we need to make some further approximations in order to optimize $L$ over it using stochastic gradient descent. To get around this, we maintain continuous surrogates $\hat{\Phi}$.

For optimizing the classification loss, we use the "straight-through" gradient estimator (Bengio et al., 2013), which provides a biased gradient estimate but has shown good results in practice. This consists of rounding the continuous surrogate to the nearest integer during training, and ignoring the rounding for purposes of backpropagation. After training, we only keep the discretized values.

In order to obtain good estimates for both the rate term and its gradient during training, we adopt a relaxation approach previously described by Ballé, Minnen, et al. (2018, appendix 6.1); the code is provided as an open source library[1]. In a nutshell, the method replaces the probability mass functions $q_i$ with a set of non-parametric continuous density functions, which are based on small ANNs. These density models are fitted to $\hat{\phi}_{j,i} + n_{j,i}$, where $n_{j,i} \sim \mathcal{U}(-\frac{1}{2}, \frac{1}{2})$ is i.i.d. uniformly distributed additive noise. This turns out to work well in practice, because the negative log likelihood of these noise-affected variates under the continuous densities approximates the self-information $I$:

$$I(\boldsymbol{\phi}) \approx \sum_{j=1}^{d} \sum_{i=1}^{\ell} -\log_2 \tilde{q}_i(\phi_{j,i} + n_{j,i}), \tag{8}$$

---

[1]https://github.com/tensorflow/compression

where $\tilde{q}_i$ denote the density functions. Once the density models are trained, the values of the probability mass functions modeling $\phi$ are derived from the substitutes $\tilde{q}_i$ and stored in a table, which is included in the model description. The parameters of $\tilde{q}_i$ are no longer needed after training.

## 2.3 MODEL PARTITIONING

A central component of our approach is partitioning the set of model parameters into groups. For the purpose of creating a model compression method, we interpret entire groups of model parameters as samples from the same learned distribution. We define a fully factorized distribution $q(\Phi) = \prod_{\phi \in \Phi} q_\phi(\phi)$, and introduce parameter sharing within the factors $q_\phi$ of the distribution that correspond to the same group, as well as within the corresponding decoders. These group assignments are fixed a priori. For instance, in fig. 2, $\widetilde{W}^1$ and $\widetilde{W}^2$ can be assumed to be samples of the same distribution, that is $q_{\widetilde{W}^1} = q_{\widetilde{W}^2}$. We also use the same parameter decoder $f_{\text{conv}}$ to decode them. Further, each of the reparameterizations $\phi$ is defined as a rank-2 tensor (a matrix), where each row corresponds to a "sample" from the learned distribution. The operations in $f$ apply the same transformation to each row (fig. 3). As an example, in $f_{\text{conv}}$, each spatial $H \times W$ matrix of filter coefficients is assumed to be a sample from the same distribution.

Our method can be applied analogously to various model partitionings. In fact, in our experiments, we vary the size of the groups, i.e., the number of parameters assumed i.i.d., depending on the total number of parameters of the model ($\Theta$). The size of the groups parameterizes a trade-off between compressibility and overhead: if groups consisted of just one scalar parameter each, compressibility would be maximal, since $q$ would degenerate (i.e., would capture the value of the parameter with certainty). However, the overhead would be maximal, since $\mathcal{F}$ and $q$ would have a large number of parameters that would need to be included in the model size (defeating the purpose of compression). On the other hand, encoding all parameters of the model with one and the same decoder and scalar distribution would minimize overhead, but may be overly restrictive by failing to capture distributional differences amongst all the parameters, and hence lead to suboptimal compressibility. We describe the group structure of each network that we use in more detail in the following section.

## 3 EXPERIMENTS

For our MNIST and CIFAR-10 experiments, we evaluate our method by applying it to four distinct image classification networks: LeNet300-100 (Lecun et al., 1998) and LeNet-5-Caffe[2] on MNIST (LeCun and Cortes, 2010), as well as VGG-16[3] (Simonyan and Zisserman, 2015) and ResNet-20 (He et al., 2016b; Zagoruyko and Komodakis, 2016) with width multiplier 4 (ResNet-20-4) on CIFAR-10 (Zagoruyko and Komodakis, 2016). For our ImageNet experiments, we evaluate our method on the ResNet-18 and ResNet-50 (He et al., 2016a) networks. We train all our models from scratch and compare them with recent state-of-the-art methods by quoting performance from their respective papers. Compared to many previous approaches, we do not initialize the network with pre-trained or pre-sparsified weights.

We found it useful to use two separate optimizers: one to optimize the variables of the probability models $\tilde{q}_i$, and one to optimize the reparameterizations $\Phi$ and variables of the parameter decoders $\Psi$. While the latter is chosen to be the same optimizer typically used for the task/architecture, the former is always Adam (Kingma and Ba, 2015) with a learning rate of $10^{-4}$. We chose to always use Adam, because the parameter updates used by Adam are independent of any scaling of the objective (when its hyper-parameter $\epsilon$ is sufficiently small). In our method, the probability model variables only get gradients from the entropy loss which is scaled by the rate penalty $\lambda$. Adam normalizes out this scale and makes the learning rate of the probability model independent of $\lambda$ and of other hyperparameters such as the model partitioning.

## 3.1 MNIST EXPERIMENTS

We apply our method to two LeNet variants: LeNet300-100 and LeNet5-Caffe and report results in table 1. We train the networks using Adam with a constant learning rate of 0.001 for 200,000

---

[2]https://github.com/BVLC/caffe/tree/master/examples/mnist
[3]http://torch.ch/blog/2015/07/30/cifar.html

iterations. To remedy some of the training noise from quantization, we maintain an exponential moving average (EMA) of the weights and evaluate using those. Note that this does not affect the quantization, as quantization is performed after the EMA variables are restored.

LeNet300-100 consists of 3 fully connected layers. We partitioned this network into three parameter groups: one for the first two fully connected layers, one for the classifier layer, and one for biases. LeNet5-Caffe consists of two $5 \times 5$ convolutional layers followed by two fully connected layers, with max pooling following each convolutional layer. We partitioned this network into four parameter groups: One for both of the convolutional layers, one for the penultimate fully connected layer, one for the final classifier layer, and one for the biases.

As evident from table 1, for the larger LeNet300-100 model, our method outperforms all the baselines while maintaining a comparable error rate. For the smaller LeNet5-Caffe model, our method is second only to Minimal Random Code Learning (Havasi et al., 2019). Note that in both of the MNIST models, the number of probability distributions $\ell = 1$ in every parameter group, including in the convolutional layers. To be precise, the $\widetilde{\boldsymbol{W}}^k$ for the convolutional weights $\boldsymbol{W}^k$ will be $H \cdot W \cdot I \cdot O \times 1$. This is a good trade-off, since the model is small to begin with, and having $\ell = 5 \cdot 5 = 25$ scalar probability models for $5 \times 5$ convolutional layers would have too much overhead.

For both of the MNIST models, we found that letting each subcomponent of $\mathcal{F}$ be a simple dimension-wise scalar affine transform (similar to $f_{\text{dense}}$ in fig. 3), was sufficient. Since each $\phi$ is quantized to integers, having a flexible scale and shift leads to flexible SQ, similar to Louizos, Reisser, et al. (2019). Due to the small size of the networks, more complex transformation functions would lead to too much overhead.

## 3.2 CIFAR-10 EXPERIMENTS

We apply our method to VGG-16 (Simonyan and Zisserman, 2015) and ResNet-20-4 (He et al., 2016b; Zagoruyko and Komodakis, 2016) and report the results in table 1. For both VGG-16 and ResNet-20-4, we use momentum of 0.9 with an initial learning rate of 0.1, and decay by 0.2 at iterations 256,000, 384,000, and 448,000 for a total of 512,000 iterations. This learning rate schedule was fixed from the beginning and was not tuned in any way other than verifying that our models' training loss had converged.

VGG-16 consists of 13 convolutional layers of size $3 \times 3$ followed by 3 fully connected layers. We split this network into four parameter groups: one for all convolutional layers and one each all fully connected layers. We did not compress biases. We found that the biases in 32-bit floating point format add up to about 20 KB, which we add to our reported numbers.

ResNet-20-4 consists of 3 ResNet groups with 3 residual blocks each. There is also an initial convolution layer and a final fully connected classification layer. We partition this network into two parameter groups: one for all convolutional layers and one for the final classification layer. We again did not compress biases and include them in our results; they add up to about 11 KB.

For convolutions in both CIFAR-10 models, $\ell = O \times I = 9$; $f_{\text{conv}}$ and $f_{\text{dense}}$ are exactly as pictured in fig. 3. To speed up training, we fixed $\boldsymbol{\psi_W}$ to a diagonal scaling matrix multiplied by the inverse real-valued discrete Fourier transform (DFT). We found that this particular choice performs much better than SQ, or than choosing a random but fixed orthogonal matrix in place of the DFT (fig. 4). From the error vs. rate plots, the benefit of reparameterization in the high compression regime is evident. VGG-16 and ResNet-20-4 both contain batch normalization (Ioffe and Szegedy, 2015) layers that include a moving average for the mean and variance. Following Havasi et al. (2019), we do not include the moving averages in our reported numbers. We do, however, include the batch normalization bias term $\beta$ and let it function as the bias for each layer ($\gamma$ is set to a constant 1).

## 3.3 IMAGENET EXPERIMENTS

For the ImageNet dataset (Russakovsky et al., 2015), we reproduce the training setup and hyperparameters from He et al. (2016a). We put all $3 \times 3$ convolutional kernels in a single parameter group, similar to in our CIFAR experiments. In the case of ResNet-50, we also group all $1 \times 1$ convolutional kernels together. We put all the remaining layers in their own groups. This gives a

| Model | Algorithm | Size | Error (Top-1) |
|---|---|---|---|
| LeNet300-100 (MNIST) | Uncompressed | 1.06 MB | 1.6% |
| | Bayesian Compression (GNJ) (Louizos, Ullrich, et al., 2017) | 18.2 KB (58x) | 1.8% |
| | Bayesian Compression (GHS) (Louizos, Ullrich, et al., 2017) | 18.0 KB (59x) | 2.0% |
| | Sparse Variational Dropout (Molchanov et al., 2017) | 9.38 KB (113x) | 1.8% |
| | Our Method (SQ) | **8.56 KB (124x)** | 1.9% |
| LeNet5-Caffe (MNIST) | Uncompressed | 1.72 MB | 0.7% |
| | Sparse Variational Dropout (Molchanov et al., 2017) | 4.71 KB (365x) | 1.0% |
| | Bayesian Compression (GHS) (Louizos, Ullrich, et al., 2017) | 2.23 KB (771x) | 1.0% |
| | Minimal Random Code Learning (Havasi et al., 2019) | **1.52 KB (1110x)** | 1.0% |
| | Our Method (SQ) | 2.84 KB (606x) | 0.9% |
| VGG-16 (CIFAR-10) | Uncompressed | 60 MB | 6.6% |
| | Bayesian Compression (Louizos, Ullrich, et al., 2017) | 525 KB (116x) | 9.2% |
| | DeepCABAC (Wiedemann, Kirchhoffer, et al., 2019) | 960 KB (62.5x) | 9.0% |
| | Minimal Random Code Learning (Havasi et al., 2019) | 417 KB (159x) | 6.6% |
| | Minimal Random Code Learning (Havasi et al., 2019) | 168 KB (452x) | 10.0% |
| | Our Method (DFT) | **101 KB (590x)** | 10.0% |
| ResNet-20-4 (CIFAR-10) | Uncompressed | 17.2 MB | 5% |
| | Our Method (SQ) | 176 KB (97x) | 10.3% |
| | Our Method (DFT) | **128 KB (134x)** | 8.8% |
| ResNet-18 (ImageNet) | Uncompressed | 46.7 MB | 30.0% |
| | AP + Coreset-S (Dubey et al., 2018) | 3.11 MB (15x) | 32.0% |
| | Our Method (SQ) | 2.78 MB (17x) | 30.0% |
| | Our Method (DFT) | **1.97 MB (24x)** | 30.0% |
| ResNet-50 (ImageNet) | Uncompressed | 102 MB | 25% |
| | AP + Coreset-S (Dubey et al., 2018) | 6.46 MB (16x) | 26.0% |
| | DeepCABAC (Wiedemann, Kirchhoffer, et al., 2019) | 6.06 MB (17x) | 25.9% |
| | Our Method (SQ) | 5.91 MB (17x) | 26.5% |
| | Our Method (DFT) | **5.49 MB (19x)** | 26.0% |

Table 1: Our compression results compared to the existing state of the art. Our method is able to achieve higher compression than previous approaches in LeNet300-100, VGG-16, and ResNet-18/50, while maintaining comparable prediction accuracy. We have reported the models that have the closest accuracy to the baselines. For the complete view of the trade-off refer to figs. 4a and 4b. For VGG-16 and ImageNet experiments, we report a median of three runs with a fixed entropy penalty. For ResNet-20-4, we report the SQ and DFT points closest to 10% error from fig. 4b. Note that the values we reproduce here for MRC are the corrected values found in the OpenReview version of the publication.

total of 4 parameter groups for ResNet-50 and 3 groups for ResNet-18. Analogously to the CIFAR experiments, we compare SQ to using random orthogonal or DFT matrices for reparameterizing the convolution kernels (fig. 4a).

## 4 DISCUSSION

Existing model compression methods are typically built on a combination of *pruning*, *quantization*, or *coding*. Pruning involves sparsifying the network either by removing individual parameters or higher level structures such as convolutional filters, layers, activations, etc. Various strategies for pruning weights include looking at the Hessian (Cun et al., 1990) or just their $\ell_p$ norm (Han et al., 2016). Srinivas and Babu (2015) focus on pruning individual units, and H. Li et al. (2017) prunes convolutional filters. Louizos, Ullrich, et al. (2017) and Molchanov et al. (2017), which we compare to in our compression experiments, also prune parts of the network. Dubey et al. (2018) describe a dimensionality reduction technique specialized for CNN architectures. Pruning is a simple approach to reduce memory requirements as well as computational complexity, but doesn't inherently tackle the problem of efficiently representing the parameters that are left. Here, we primarily focus on the latter: given a model architecture and a task, we're interested in finding a set of parameters which can be described in a compact form and yield good prediction accuracy. Our work is largely orthogonal to the pruning literature, and could be combined if reducing the number of units is desired.

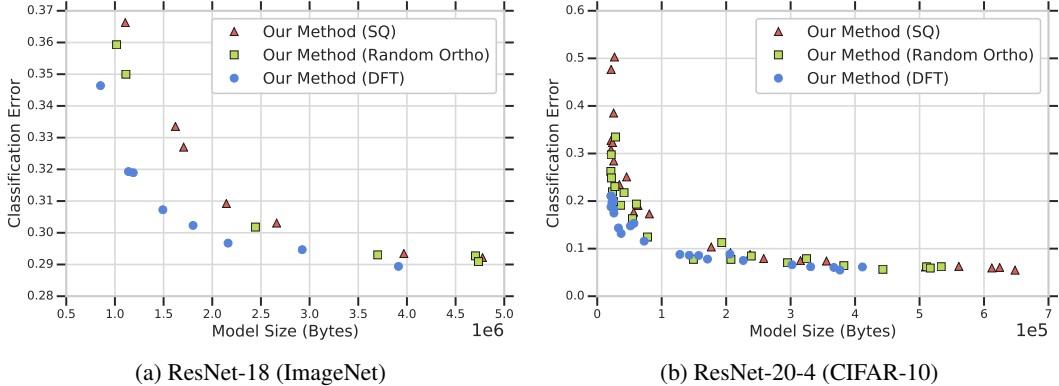

(a) ResNet-18 (ImageNet)    (b) ResNet-20-4 (CIFAR-10)

Figure 4: Error vs. rate plot for ResNet-18 on ImageNet and ResNet-20-4 on CIFAR-10 using SQ, DFT transform, and random but fixed orthogonal matrices. The DFT is clearly beneficial in comparison to the other two transforms. All experiments were trained with the same hyper-parameters (including the set of entropy penalties), only differing in the transformation matrix.

Quantization involves restricting the parameters to a small discrete set of values. There is work in binarizing or ternarizing networks (Courbariaux et al., 2015; F. Li et al., 2016; Zhou et al., 2018) via either straight-through gradient approximation (Bengio et al., 2013) or stochastic rounding (Gupta et al., 2015). Recently, Louizos, Reisser, et al. (2019) introduced a new differentiable quantization procedure that relaxes quantization. We use the straight-through heuristic, but could possibly use other stochastic approaches to improve our methods. While most of these works focus on uniform quantization, Baskin et al. (2018) also extend to non-uniform quantization, which our generalized transformation function amounts to. Han et al. (2016) and Ullrich et al. (2017) share weights and quantize by clustering, Chen, J. Wilson, et al. (2015) randomly enforce weight sharing, and thus effectively perform VQ with a pre-determined assignment of parameters to representers. Other works also make the observation that representing weights in the frequency domain helps compression; Chen, J. T. Wilson, et al. (2016) randomly enforce weight sharing in the frequency domain and Wang et al. (2016) use K-means clustering in the frequency domain.

Coding (entropy coding, or Shannon-style compression) methods produce a bit sequence that can allow convenient storage or transmission of a trained model. This generally involves quantization as a first step, followed by methods such as Huffman coding (Huffman, 1952), arithmetic coding (Rissanen and Langdon, 1981), etc. Entropy coding methods exploit a known probabilistic structure of the data to produce optimized binary sequences whose length ideally closely approximates the cross entropy of the data under the probability model. In many cases, authors represent the quantized values directly as binary numbers with few digits (Courbariaux et al., 2015; F. Li et al., 2016; Louizos, Reisser, et al., 2019), which effectively leaves the probability distribution over the values unexploited for minimizing model size; others do exploit it (Han et al., 2016). Wiedemann, Marban, et al. (2018) formulate model compression with an entropy constraint, but use (non-reparameterized) scalar quantization. Their model significantly underperforms all the state-of-the-art models that we compare with (table 1). Some recent work has claimed improved compression performance by skipping quantization altogether (Havasi et al., 2019). Our work focuses on coding with quantization.

Han et al. (2016) defined their method using a four-stage training process: 1. training the original network, 2. pruning and re-training, 3. quantization and re-training, and 4. entropy coding. This approach has influenced many follow-up publications. In the same vein, many current high-performing methods have significant complexity in implementation or require a multi-stage training process. Havasi et al. (2019) requires several stages of training and retraining while keeping parts of the network fixed. Wiedemann, Kirchhoffer, et al. (2019) require pre-sparsification of the network, which is computationally expensive, and use a more complex (context-adaptive) variant of arithmetic coding which may be affected by MPEG patents. These complexities can prevent methods from scaling to larger architectures or decrease their practical usability. In contrast, our method requires only a single training stage followed by a royalty-free version of arithmetic coding. In

addition, our code is publicly available[4].

Our method has parallels to recent work in learned image compression (Ballé, Laparra, et al., 2017; Theis et al., 2017) that uses end-to-end trained deep models for significant performance improvements in lossy image compression. These models operate in an autoencoder framework, where scalar quantization is applied in the latent space. Our method can be viewed as having just a decoder that is used to transform the latent representation into the model parameters, but no encoder.

## 5 CONCLUSION

We describe a simple model compression method built on two ingredients: joint (i.e., end-to-end) optimization of compressibility and task performance in only a single training stage, and reparameterization of model parameters, which increases the flexibility of the representation over scalar quantization, and is applicable to arbitrary network architectures. We demonstrate that state-of-the-art model compression performance can be achieved with this simple framework, outperforming methods that rely on complex, multi-stage training procedures. Due to its simplicity, the approach is particularly suitable for larger models, such as VGG and especially ResNets. In future work, we may consider the potential benefits of even more flexible (deeper) parameter decoders.

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
