# OpenReview forum: "Scalable Model Compression by Entropy Penalized Reparameterization"
_ICLR.cc/2020/Conference — Accept (Poster)_

### Official Review · AnonReviewer2 · 2019-10-22
**Official Blind Review #2**

**Rating:** 6

**Review:**

The authors propose a novel, information theoretic approach to learning compressed neural networks, whereby model parameters $\theta$ are expressed in an (approximately discrete) latent space as $\theta = f_{\psi}(\phi)$. During training, "reparameterizations" $\Phi$ are treated as draws from a generative distribution $q(\phi)$. As per the source coding theorem, penalizing the entropy of $q$ (more accurately, of $\hat{q}$) directly promotes compressible choices of $\Phi$ and, therefore, of $\Theta$. After training, arithmetic coding of $q$ is used to compress $\Phi$. Empirical results on a suite of common benchmarks demonstrates the proposed algorithm gracefully trades off between model performance and size.


Frequency Domain Interpretation:
  Here, I am operating off of the assumptions that, in Sect 3.2 paragraph 3, $\phi_{w}$ should be $\psi_{w}$. If so, can "Our Method (DFT)" also be interpreted as frequency domain SQ (of convolutional filters)? When trained on natural images, CNN filters are typically "smooth" (see any visualization thereof) and this smoothness translates as a prior of sorts on $\Phi_{w}$. This insight was previously explored in [1, 2] for purposes of compressing CNNs. Does evidence of this manifest in your experiments? For example, do the empirical distributions of high- and low-frequency components of $\phi_{w}$ differ?


Feedback:
  The authors propose a general framework where in (seemingly) arbitrary decoders $f$ can be used for purposes of "reparameterization". It is therefore somewhat disappointing that, in practice, $f$ is restricted to affine transformations and, in some cases, even fixed. Memory issues notwithstanding, it is unclear how well complex decoders $f$ could be jointly learned with (surrogate) encodings $\tilde{\Phi}$ and density models $\tilde{q}$.

  The lack of comparison between the proposed method's test-time throughput and that of its baselines leaves open questions regarding potential real-world use cases. Much of the time, methods for compressing neural networks divide their attention between both memory footprint and prediction speed. Outright ignoring this aspect of the problem seems odd. As a reviewer, I would much rather see the authors take a proactive approach in addressing this issue.


Questions:
  - What do results look like if convolutional decoders $f_{\text{conv}}$ are jointly learned during training?
  - How sensitive is the algorithm to different groups assignments? Can assignments be reliably made by simply looking at the architecture?


Minor Comments:
  - The general notation of the paper is cumbersome at times, can $\phi$ to be changed to, e.g., $\tilde{\theta}$? Similarly, can notation such as $\theta_{k, W}$ be simplified as $W^{k}$?

  - Section 2 jumps around a bit. Consider trying to order things "chronologically" such that training comes before compression? By the same token, details regarding "Model Partitioning" could be move to the end of the section such that the preceding material just refers to $\theta$ more abstractly?

  - Some additional details regarding training (esp. discretization and model distributions $\tilde{q}$) would be appreciated.


[1] "Compressing convolutional neural networks in the frequency domain", Chen et al. 2016
[2] "CNNpack: Packing convolutional neural networks in the frequency domain", Wang et al. 2016

**Experience Assessment:**

I have published one or two papers in this area.

**Review Assessment: Checking Correctness Of Derivations And Theory:**

I assessed the sensibility of the derivations and theory.

**Review Assessment: Checking Correctness Of Experiments:**

I assessed the sensibility of the experiments.

**Review Assessment: Thoroughness In Paper Reading:**

I read the paper at least twice and used my best judgement in assessing the paper.

---

> ### Author Response · Authors · 2019-11-12
> **Response to Blind Review #2**
>
> Thank you for taking the time to review our paper! We will address your points one by one.
>
> Frequency domain interpretation: Thank you for making us aware of these papers. Note that while some of the intuitions in these papers are shared with our choice of f_conv, we present a more general framework. Indeed, we did observe a significant difference between the empirical distributions of the high vs. low frequency components when training with the DFT transformation as learned via end-to-end optimization. A key property of our method is that since each column of \Phi is treated as a separate probability distribution, certain columns can be collapsed to a delta distribution (leading to 0 entropy and thus very high compression). We noticed that for some of our experiments using the DFT transformation, the columns corresponding to the high frequency components had collapsed, and the model had learned to just use the low frequency components and thus achieve very high compression rates. We added references to the papers you mentioned in the discussion section, and will look into adding a more in depth discussion for the camera ready.
>
> Jointly learning convolutional decoders: This is certainly an area we wish to explore further. We found that jointly learning an affine f_conv, rather than fixing it to DFT, provided even better results than DFT in the high compression regime, but did not optimize well in the low compression regime. With some tuning of learning rate, initialization, etc., we think it is possible to get better results across all regimes, but we decided to stick with the DFT parameterization due to simplicity being a key goal of our model for practical application. We plan on exploring deeper transformations as well. We hypothesize that invertible models or flow-based models for the convolutional decoder could lead to further compression benefits.
>
> Grouping: The groups were purely determined by looking at architecture, and not tuned in any way. We used the following assignments for all networks: 1 group for all 3x3 convolutions, 1 group for all 1x1 convolutions, 1 group for each fully connected layer. Both of the 5x5 convolutional layers in LeNet-5 were also in one group, even though they were treated as fully connected layers (since a linear decoder would have too much size overhead; this is mentioned in the paper). It is possible that tuning the group assignments might lead to further benefits.
>
> Test-time throughput: We agree that test-time throughput is an important metric when using networks in practice, and many sparsification and quantization techniques lead to significant benefits on this front. Our method is orthogonal to sparsification techniques, and could be combined if some amount of throughput improvement is desired. While pruning techniques admit relatively simple ways of quantifying computational benefit (as the runtime will approximately go down linearly with each removed unit), quantifying the overhead of parameter decoding is more difficult, since the relative computational complexity between arithmetic decoding and, say, computing a convolution, highly depends on the platform and the implementation. (Note that this also applies to other baselines we compare to that go beyond pruning.) As an example, our current implementation of arithmetic coding is confined to the CPU, while the rest of the model runs on a GPU. Once decompressed, our models have the same runtime as the corresponding uncompressed baseline. In a typical workstation setup, the parameter decompression runs in parallel with the computations on the GPU (since the decompression doesn’t depend on the input data), thus requiring essentially no runtime overhead. Due to these implementation dependencies, we think answering your question accurately for any given platform would require a substantial amount of implementation work which would exceed the scope of this paper. In a setting where decompression happens very infrequently, such as when memory, network bandwidth, or storage capacity is limited (i.e. for long-term storage, deployment, or a pipelining scenario on low-memory devices), however, the benefits of our method should be clear.
>
> Notation: We followed your suggestion to simplify the notation. We think it is clearer now.
>
> Chronology: Thank you for that suggestion. We will consider polishing that part of the text for the camera-ready version.
>
> Training: We only briefly review the discretization and probability modeling here, since it isn’t part of our contribution, and we wanted to keep the paper length manageable. The details of discretization and maintaining continuous surrogates were first described in https://arxiv.org/abs/1607.05006, while the parametric density model we use is described in https://arxiv.org/abs/1802.01436, appendix 6.1. We will make sure that this is mentioned in the final version of the paper.

---

### Official Review · AnonReviewer3 · 2019-10-23
**Official Blind Review #3**

**Rating:** 8

**Review:**

The paper presents a compression algorithm for neural networks. It uses a linear projection to map the weights and biases to a latent space where they are quantized using a learnt coding distribution. The method is simple yet it shows strong performance of a variety of compression benchmarks.

Originality: The core idea is related to (Balle et al. 2018), but the paper puts its own twist to it with the projections and applies it to model compression. It is certainly an interesting direction to explore.

Presentation: The paper is well written. It is easy to read and understand. It properly cites prior works and contains all the technical details. I appreciate that the authors fit the paper into the recommended 8 pages.

Impact: The method shows very strong performance in terms of compression ratio, but its unclear whether the compressed model can be used to speed up inference. Currently the main use case would be saving storage/bandwidth.

Questions:
- In section 2.2, the paper talks about the form of the coding distribution. It has d components and l dimensions. How is d determined?
- Section 2.1, How are the weights grouped for coding? is every filter its own group? If the experiments use different groups for different models, how is it decided which model uses which approach?
- \phi was fixed for the CIFAR10/ImageNet experiments. Could you provide further insight into why this choice was made?

Assessment:
While the idea is not groundbreaking, it is very well presented and evaluated and shows strong performance.

**Experience Assessment:**

I have published one or two papers in this area.

**Review Assessment: Checking Correctness Of Derivations And Theory:**

I assessed the sensibility of the derivations and theory.

**Review Assessment: Checking Correctness Of Experiments:**

I assessed the sensibility of the experiments.

**Review Assessment: Thoroughness In Paper Reading:**

I read the paper at least twice and used my best judgement in assessing the paper.

---

> ### Author Response · Authors · 2019-11-12
> **Response to Blind Review #3**
>
> Thank you for taking the time to review our paper! We will address your points one by one.
>
> How is d determined? d is the number of spatial kernels in the kernel tensor for f_conv (i.e., IxO), and l is the latent dimensionality, which we choose identical to the number of dimensions of the parameter, (i.e. HxW=9 for a 3x3 spatial kernel).
>
> Grouping: The groups were purely determined by looking at architecture, and all experiments use the same grouping scheme. Furthermore, the groups were not tuned in any way. We used the following assignments for all networks: 1 group for all 3x3 convolutions, 1 group for all 1x1 convolutions, 1 group for each fully connected layer. Both of the 5x5 convolutional layers in LeNet-5 were also in one group, even though they were treated as fully connected layers (since a linear decoder would have too much size overhead; this is mentioned in the paper).
>
> Fixed decoder: While we found that jointly learning an affine f_conv, rather than fixing it to DFT, provided even better results than DFT in the high compression regime, it did not optimize well in the low compression regime. With some tuning of learning rate, initialization, etc., we think it is possible to get better results across all regimes with a learned decoder, but we decided to stick with the DFT parameterization due to simplicity being a key goal of our model for practical application.

---

### Official Review · AnonReviewer4 · 2019-11-02
**Official Blind Review #4**

**Rating:** 6

**Review:**

The paper introduces an end-to-end method for the neural network compression, which is based on compressing reparametrized forms of model parameters (weights and biases). Shannon-style entropy coder is applied to the reparametrizations.

Reparameterized forms are tensors stored in the compressed format. During inference, they are uncompressed into integer tensors and transformed via parameter decoders into weights/biases of convolutional and dense layers.

During training, model parameters are manually partitioned into groups, and within a group they are considered as samples from the same learned distribution. Similarly, parameter sharing is introduced among the corresponding parameter decoders.
Loss function, which is minimized during training is a sum of rate loss (self-information of all reparametrizations) and cross-entropy classification loss under reparametrization. A trade-off between compressed model size and model accuracy can be explored by varying a constant before the rate loss.

During optimization, several tricks are applied. “Straight-through” gradient estimator is used to optimize the loss function over discrete-valued reparametrizations by means of stochastic gradient descent. Relaxation is used to obtain good estimates for both the rate term and its gradient during training.

The proposed idea is well-founded and intuitive. The proposed method is extensively evaluated on different classification network architectures and datasets. It provides good compression while retaining a significant portion of accuracy.

In the experiments, It'd be interesting to see a comparison on more efficient networks like MobileNets, ShuffleNets on ImageNet dataset. Also, I wonder whether under the same compression rate the proposed method outperforms DeepCompression (Han et al., 2015) in terms of accuracy? (for example, for LeNets and VGG-16)


**Experience Assessment:**

I have published one or two papers in this area.

**Review Assessment: Checking Correctness Of Derivations And Theory:**

I assessed the sensibility of the derivations and theory.

**Review Assessment: Checking Correctness Of Experiments:**

I assessed the sensibility of the experiments.

**Review Assessment: Thoroughness In Paper Reading:**

I read the paper thoroughly.

---

> ### Author Response · Authors · 2019-11-12
> **Response to Blind Review #4**
>
> Thank you for your review, and the suggestions. While we believe to already have covered a wide array of network architectures supporting our claim that our method generalizes well, in particular to larger networks, we agree that experiments on MobileNets/ShuffleNets would be interesting. We will make an effort to run further experiments and include these numbers in the final paper. However, our priority will be on polishing the source code of our method, in order to make it public and enable others to apply the method to even more architectures.

---

### Author Response · Authors · 2019-11-12
**New revision uploaded**

Dear reviewers,

thank you for your valuable suggestions. We uploaded a new version of the paper with some of the changes we discuss in our individual responses below. In particular, we simplified the notation, and we now discuss further references, as suggested. We also took the liberty to improve some of the figures. Due to time constraints, we didn't make all changes we'd want to make yet, but we intend to submit at least one more revision before the final paper is due.

Thank you,
the authors.

---

### Decision · Program_Chairs · 2019-12-19

**Decision:**

Accept (Poster)

**Comment:**

The paper describes a simple method for neural network compression by applying Shannon-type encoding. This is a fresh and nice idea, as noted by reviewers. A disadvantage is that the architectures on ImageNet are not the most efficient ones. Also, the review misses several important works on low-rank factorization of weights for the compression (Lebedev et. al, Novikov et. al).   But overall, a good paper.